# Medication Trends for Age-Related Macular Degeneration

**DOI:** 10.3390/ijms222111837

**Published:** 2021-10-31

**Authors:** Yeon-Kyoung Cho, Dae-Hun Park, In-Chul Jeon

**Affiliations:** 1College of Health and Welfare, Dongshin University, Naju 58245, Jeonnam, Korea; yktender@naver.com; 2College of Korean Medicine, Dongshin University, Naju 58245, Jeonnam, Korea

**Keywords:** age-related macular degeneration (AMD), medication, device-based therapy, anti-inflammatory drugs, anti-VEGFs, natural products

## Abstract

Age-related macular degeneration (AMD) is central vision loss with aging, was the fourth main cause of blindness in 2015, and has many risk factors, such as cataract surgery, cigarette smoking, family history, hypertension, obesity, long-term smart device usage, etc. AMD is classified into three categories: normal AMD, early AMD, and late AMD, based on angiogenesis in the retina, and can be determined by bis-retinoid *N*-retinyl-*N*-retinylidene ethanolamine (A2E)-epoxides from the reaction of A2E and blue light. During the reaction of A2E and blue light, reactive oxygen species (ROS) are synthesized, which gather inflammatory factors, induce carbonyl stress, and finally stimulate the death of retinal pigment epitheliums (RPEs). There are several medications for AMD, such as device-based therapy, anti-inflammatory drugs, anti-VEGFs, and natural products. For device-based therapy, two methods are used: prophylactic laser therapy (photocoagulation laser therapy) and photodynamic therapy. Anti-inflammatory drugs consist of corticosteroids and non-steroidal anti-inflammatory drugs (NSAIDs). Anti-VEGFs are classified antibodies for VEGF, aptamer, soluble receptor, VEGF receptor-1 and -2 antibody, and VEGF receptor tyrosine kinase inhibitor. Finally, additional AMD drug candidates are derived from natural products. For each medication, there are several and severe adverse effects, but natural products have a potency as AMD drugs, as they have been used as culinary materials and/or traditional medicines for a long time. Their major application route is oral administration, and they can be combined with device-based therapy, anti-inflammatory drugs, and anti-VEGFs. In general, AMD drug candidates from natural products are more effective at treating early and intermediate AMD. However, further study is needed to evaluate their efficacy and to investigate their therapeutic mechanisms.

## 1. Introduction

### 1.1. Definition

Age-related macular degeneration (AMD) is related to aging and is a disease that causes problems in the central region of the retina, also known as the macula [1]. AMD patients experience several physiological changes, such as the loss of central vision, drusen, retinal pigmentary changes, angiogenesis, etc., and according to angiogenesis in the retina, it can be classified as dry AMD or wet AMD, with the percentage of wet AMD cases reported to be 10–15% of the total AMD cases [2].

### 1.2. Health Problems, Statistics, and Risk Factors

In 2015, AMD was the fourth most common cause of blindness, and among patients with moderate to severe vision impairment, it was the third most common cause [3]. Among those aged 80 years and above, 66% have several signs of AMD [4,5], and 3% of those over 65 years old suffer visual problems caused by AMD [6]. According to Zou’s study on the global burden of AMD from 1990 to 2017, special groups can significantly suffer from AMD, such as females, the elderly, those from developing countries, and low-educated persons, etc., and from 1990 to 2017, AMD contributed to a doubling of the number of people living a low quality of life [7]. An even more serious consideration is that the number of global AMD patients has been predicted to rise from 196 million in 2020 to 288 million in 2040 [8].

There are many risk factors associated with AMD, such as tobacco smoking, age increasing, experiences of cataract surgery, family history, high body mass index (BMI) values, cardiovascular disease, hypertension, plasma fibrinogen, obesity, atherosclerosis, high-density lipoprotein cholesterol (HDC-C), etc. [9,10,11,12]. Furthermore, as smart device use grows, ophthalmological problems can come to the fore, especially the probability of AMD occurrence, which might be caused by increases in smartphones’ blue light [13,14]; for this reason, a reduction in the smartphone usage of children and juveniles has been recommended [15].

### 1.3. Classification of Current Medication

Currently, there are several types of AMD medication (Table 1), including device-based treatment (prophylactic laser or photocoagulation therapy, photodynamic therapy), anti-inflammatory drug treatment (corticosteroids, nonsteroidal anti-inflammatory drugs (NSAIDs)), anti-vascular endothelial growth factor (anti-VEGF) intravitreal injection, nutritional therapy (antioxidants, ω-3 fatty acids, lutein, zeaxanthin, long-chain polyunsaturated fatty acids, plant extracts, etc.), etc. We discuss here the current AMD medications and their advantages and disadvantages, and consider how to prevent or improve the symptoms of intermediate AMD. 

## 2. Age-Related Macular Degeneration (AMD)

### 2.1. Category

AMD can be classified into two types: early (dry) AMD and late (wet, exudative) AMD, depending on the angiogenesis or lack thereof in the retina. In order to harmonize the classification of AMD in 2014, researchers from three continents reported that according to the sizes and types of areas of drusen, early AMD can be categorized into three types: mild early AMD, moderate early AMD, and severe early AMD. Late AMD can be classified as geographic atrophy or neovascular AMD and can be defined by the occurrence of atrophy, the formation of new blood vessels in the retina, pigment epithelial detachment and/or retinal detachment, etc. [16]. 

### 2.2. Pathogenic Factors

With advancing years, lipofuscin, also called aging pigments, accumulates in several cells after mitosis, such as neurons, cardiac cells, and retinal pigment epithelium (RPE), and bis-retinoid *N*-retinyl-*N*-retinylidene ethanolamine (A2E) are two of the main causes of lipofuscin in our body [17]. A2E accumulation in our body is an aging phenomenon, and although the mechanism behind it is unclear, it is vulnerable to oxidative damage [18]. A2E is very fickle, and it depends on the wavelength of light and the A2E oxidation form induced by blue light (480 ± 20 nm) [19]; A2E-epoxides induce cell damage and then finally stimulate cell death [20]. In RPE cells, A2E treatment has been reported to increase the level of VEGF mRNA and protein but did not affect the expression of vascular endothelial growth factor receptor 1 (VEGFR1) or VEGFR2. However, blue light exposure in A2E-loaded cells stimulated VEGFR1 but not the levels of VEGF mRNA and protein [21]. Blue light is in the special range (400–500 nm) of visible light, and in 1978, Ham and his colleagues reported that short wavelength light—i.e., blue light, not temperature—induced photochemical damage in rhesus macaques’ retinal pigment epithelium [22].

### 2.3. Pathogenesis

Oxidation is a chemical reaction that involves both the loss and simultaneous reduction in electrons in molecules [23]. Oxidation both positively and negatively affects bio-organisms: it beneficially affects them through eradicating foreign pathogenic microorganisms and modulating the function of enzymes and transcription factors via phosphorylation or dephosphorylation [24], but it damages them through breaking their cell membranes and cellular organelles and then finally inducing many chronic diseases [25,26,27]. Inflammation is the result of bio-organisms’ defense mechanisms; in innate immunity, many inflammatory cells, such as eosinophils, basophils, neutrophils, monocytes, lymphocytes, etc., accumulate in the infestation site to eliminate foreign bodies via chemotaxis [28], but if a given immune system cannot overcome an invasion of foreign bodies, inflammatory products such as enzymes (elastase, collagenase, etc.) and immune mediators (cytokines, chemokines, etc.) finally stimulate cellular damage and cell death. Inflammatory cells stimulate the release of reactive species at the inflammation site; consequently, the level of reactive species increases [29,30]. On the other hand, reactive species stimulate the release of pro-inflammatory cytokines, such as TGF-β, IL-1, etc., through activating transcription factors such as NF-κB, AP-1, and HIF-1α [31]. Carbonyl stress results from proteins that have been changed to carbonyl compounds and autoxidized by carbohydrates, lipids, or amino acids. Carbonyl compounds are called advanced glycosylation end products (AGEs). There are carbohydrate-derived AGEs, such as 3-deoxyglucosone, o-arabinose, and glyoxalate; polyunsaturated fatty acid-derived ones, such as malondialdehyde and hydroxyinonenal; an ascorbic acid-derived one, dehydroascorbate; and amino acid-derived ones, such as acrolein and 2-hydroxyipropanal [32]. 

In RPE apoptosis and oxidative stress, carbonyl stress and inflammation have a very important cause-and-effect relationship. Compared with the other tissues in the body, the level of PUFA in the RPE is very high, and RPE cells are more vulnerable to oxidative stress and carbonyl stress [33]. Especially in AMD patients, the levels of ROS such as nitrotyrosine, H_2_O_2_, Cu/Zn-superoxide dismutase (SOD1), and Mn-superoxide dismutase (SOD2) (but not NO [34,35,36,37,38]); AGEs such as malondialdehyde (MDA), 4-hydroxynonenal (4-HNE), malonaldehyde-*bis*-dimethyl acetal, 4-hydroxyhexenal (4-HHE), protein carbonyl content [38,39,40,41,42]; inflammatory cytokines such as IL-1β and IL-2, IL-6, IL-18, TNF-α via NLRP3 [39,41,43] are significantly increased, and eventually they induce RPE apoptosis via the activation of calpain-1, caspase-3, and caspase-9 and the modulation of Fas/Fas ligand (FasL), NF-κB activation, and Bax and Bad [33,44,45,46].

## 3. Device-Based Treatment

### 3.1. Prophylactic Laser Therapy (Photocoagulation Laser Therapy)

Bruch’s membrane, which supports the blood stream by stimulating the retina’s functions from the choroid layer, is located close to the RPE, but with advancing years, drusen, which are lipid-rich deposits (lipofuscin), not only accumulate between the RPE and Bruch’s membrane but also grow in size. Then, if the RPE cannot be supplied blood from the choroid layer, it faces death (apoptosis) [47,48]. Drusen are a hallmark for comparing the grades of AMD, and according to their size and distribution, AMD, including three early AMD stages, can be classified into five categories: no AMD, mild early AMD, moderate early AMD, severe early AMD, and late AMD [16] (Table 2). 

Prophylactic laser therapy is similar to photocoagulation therapy. They both involve resolving drusen using laser treatment. The mechanism of drusen resolution by laser treatment is very simple, as laser beams have enough energy (heat) to burn drusen (Figure 1). 

In 1973, the first study results on photocoagulation effects on the eyes depending on laser usage were reported, and according to the results of the study, argon laser irradiation could resolve the drusen in the eye [49]. However, if thermal lasers can be applied to drusen resolution, their heat could induce the death of RPEs near the irradiated drusen [50]. Frennesson and Nilsson [51] reported the beneficial effects of low power photocoagulation laser therapy, which affects the death of RPEs less than the high power therapy. Subthreshold diode laser treatment has been tried for suppressing AMD progress, but its application has not been well established [52]. In order to decrease the thermal effects that induce RPE death and to increase drusen resolution, prophylactic laser therapy continues to be developed. Recently, in order to avoid subordinate damage, such as RPE death, nanosecond pulse laser therapy has been developed [53]. The nanosecond pulse laser method can be precisely applied to drusen, and it has been found to be capable of resolving drusen only. The prophylactic laser therapeutics are summarized in Table 3. 

### 3.2. Photodynamic Therapy

Photomedicine can be defined as medical processes that use light for the diagnosis of and treatment of disease [54], and photochemotherapy can be classified into two categories, depending on the source of photosensitizers. One category is endogenous photosensitizers, which can absorb the light in the human body, and the other category is exogenous photosensitizers, which should be injected into the patient to accelerate their consumption of light from a medical device. The mechanisms of photodynamic therapy can be classified into three categories: anti-angiogenesis effects caused by thrombosis and stasis, anti-cancer effects caused by oxidative stress enhancement, and immune system modulation effects [55,56,57] (Figure 2). 

There are many records of the use of sunlight for healing patients [58], and in the 19th century, Finsen tried to eliminate smallpox and lupus vulgaris using sunlight’s bactericidal effect [57]. In 1903, the first clinical trial was conducted for curing cancer, psoriasis, lupus vulgaris, etc., using topical eosin under sunlight and arc irradiation, but this study was withdrawn because of pain and scar formation [59]. In 1948, using red fluorescent light tumor analysis and therapy, a study was conducted [60], and after the 1970s, hematoporphyrin derivatives (HpD), especially Photofrin^®^, were widely used for curing cancers, such as lung and esophagus cancers [61].

In order to destroy choroidal neovascularization (CNV) in AMD patients, liposomal verteporfin is widely used as a photosensitizer for its photodynamic therapeutic effects; patients should be irradiated by 689 nm light for 5 minutes after liposomal verteporfin intravenous injection [57]. However, during this therapy, an adverse effect that can damage the normal choriocapillaries has been reported; there have been many trials since to develop new photosensitizers to reduce this adverse effect [62,63,64].

## 4. Anti-Inflammatory Drugs

Drusen is one of the hallmarks for diagnosing AMD, as they are made by the destruction of RPE cells with advancing years, and the inflammation is closely related to AMD progress, as that produces drusen [65]. It is important to suppress inflammation progress to inhibit the severity of AMD (late AMD), and as shown Figure 3, there is an anti-inflammatory pathway caused by corticosteroid drugs and nonsteroidal anti-inflammatory drugs (NSAIDs).

In 2008, an epidemiological study on the relation between anti-inflammatory drug use and AMD occurrence was conducted based on 614 patients (the average age and standard deviation, 72.9 ± 6.8) and 4,526 normal veterans (the average age and standard deviation, 73.2 ± 6.7) from 1997 to 2001 [66]. According to the study’s results, anti-inflammatory drug use suppressed the occurrent risk of AMD. As the macrophage-derived proinflammatory cytokines such as tumor necrosis factor-α (TNF-α) and interleukin-1 (IL-1) especially increase when Bruch’s membrane is destroyed, and as they stimulate the increment of vascular endothelial cells, anti-inflammatory drugs can effectively suppress AMD progress [67]. In Table 4, anti-inflammatory drugs that are used in AMD treatment are summarized.

### 4.1. Corticosteroid Drugs

Corticosteroid drugs can perfectly inhibit the inflammation occurrence pathway, as they significantly control the arachidonic acid synthesis from plasma membrane breakage using phospholipase A_2_ [78] (Figure 3). Topical nanodispersion of dexamethasone and artemisinin on the eyeball effectively control CNV in AMD progress [68], and triple therapy using dexamethasone, bevacizumab, and verteporfin with photodynamic therapy is one of the anti-vascularization therapies used for AMD [69]. Triamcinolone acetonide (TA) has been widely used as a drug for macular edema and uveitis, as its effective duration is longer than that of dexamethasone [70], and combined intravitreal triamcinolone acetonide and bevacizumab injection has provided treatment to AMD patients who failed to be treated by the intravitreal injection of bevacizumab alone [71]. Spironolactone is a mineralocorticoid receptor antagonist that can be orally administered, and it is used to effectively suppress CNV in patients refractory to intravitreal anti-VEGF injections [72]. 

### 4.2. Nonsteroidal Anti-Inflammatory Drugs (NSAIDs)

Nonsteroidal anti-inflammatory drugs (NSAIDs) inhibit the synthesis of prostaglandins (PGEs) via inactivating cycloxygenase-1 (COX-1) or cycloxygenase-2 (COX-2), and they have been used as anti-inflammatory and antipyretic agents and analgesic drugs [78] (Figure 3). They are used as topical applicants for various ophthalmological purposes—for example, as an inflammation suppressor against allergic conjunctivitis and keratitis, a down-regulator for cystoid macular edema, a contractor for cataract surgery, etc. [79]. Aspirin is a famous NSAID, but although a low aspirin dose has been found to not affect AMD occurrence [73,74], a high aspirin dose might decrease AMD prevalence [75]. Nepafenac is a potent NSAID prodrug that has higher vascular permeability, longer inhibition of PGE synthesis, and better corneal penetration than others [76]. Intravitreal diclofenac and ketorolac injections have been found to effectively control inflammation in a lipopolysaccharide (LPS)-induced ocular inflammation rabbit model [77].

## 5. Anti-Vascular Endothelial Growth Factor (Anti-VEGF)

Vascularization is one of the most important aspects of AMD progress and severity, and anti-CNV therapy is an important strategy for controlling AMD. VEGF is the first inducer to be found that can stimulate sprouting angiogenesis from vascular endothelial cells and the permeability of vessels [80], and it is synthesized in ganglion, Muller, and RPE cells [81], as it is much more selective to vascular endothelial cells [82]. There are several VEGFs, including VEGF-A, VEGF-B, VEGF-C, VEGF-D, VEGF-E, and placental growth factor (PGF), and the molecular weight range of VEGFs is between 35 and 45 kd homodimers [82]. In particular, the hypoxic condition in the retina promotes VEGF excretion, and released VEGF induces pathologic angiogenesis, called CNV in AMD. Then, leaked fluid from them leads to blindness by increasing the intraocular pressure via edema, exudation, and hemorrhage in the retina area [83].

It is possible to classify the inhibitors into six groups: (1) VEGF antibodies (humanized ones, such as bevacizumab (Avastin^TM^), or fragment of humanized ones, such as ranibizumab (Lucentis^TM^)); (2) aptamers (oligonucleotides such as Pegaptanib^TM^); (3) soluble receptors (aflibercept (Eylea^TM^)); (4) anti-VEGF-R1; (5) anti-VEGF-R2; and (6) VEGFR tyrosine kinase inhibitors, such as imatinib mesylate, sorafenib, sunitinib, and vatalanib in CNV (Figure 4). 

It is difficult to cure wet and exudate AMD, but recently, various drugs for inhibiting CNV have been developed [82,84,85,86]. The most developed anti-VEGF drugs directly control VEGF activation, inhibiting the VEFG function for such a long time that they only need to be administered every 2 months [87]. In Table 5, the currently used anti-VEGF drugs are summarized. Humanized antibodies for VEGF, such as bevacizumab (Avastin^TM^; Genentech, South San Francisco, CA, USA), which is a monoclonal full-length antibody, and ranibizumab (Lucentis^TM^; Genetech), which is a monoclonal fragment (Fab) antibody, bind all active forms of VEGF [86,88]. In the RPE, two active forms of VEGF-A exist in high quantities: VEGF_165_ and VEGF_121_ [89], and VEGF aptamers such as pegaptinib (Macugen^TM^; Pfizer, New York, NY, USA) only bind VEGF_165_, inhibiting VEGF_165_ activation [90]. Aflibercept (Eylea^TM^; Regeneron Pharmaceuticals Inc., Tarrytown, NY, USA) is a soluble VEGF receptor that binds to all circulating VEGFs, such as VEGF-A, -B, -C, -D, -E, and PGF, and then traps them in order to inactivate them, similar to VEGF traps [91,92].

Recently, there have been many reports on VEGF receptor inhibitors and VEGF tyrosine kinase inhibitors, and, to date, many studies have been conducted to confirm the possibility of AMD treatment, with some exceptions. However, there are several VEGF receptor inhibitors, such as axitinib (Inlyta^®^; Pfizer, Inc., New York, NY, USA) [93,94], cabozntinib (Cabometyx^®^; Exelixis Pharmaceuticals, Inc., Alameda, CA, USA) [95], sorafenib (Nexavar^®^; LC Laboratories, Woburn, MA, USA) [96,97], SU11248/sunitinib (Sutent^®^; Pfizer, Inc., New York, NY, USA) [98], and VEGF receptor tyrosine kinase inhibitors, such as axitinib (Inlyta^®^) [93,94], cabozntinib (Cabometyx^®^) [95], sorafenib (Nexavar^®^) [96,97], and SU11248/sunitinib (Sutent^®^; Pfizer, Inc., New York, NY, USA) [98]. Furthermore, a past cancer study found that axitinib (Inlyta^®^) acts on all VEGF receptor inhibitors and on VEGF receptor tyrosine kinase inhibitor [93], and Kang et al. illuminated its anti-angiogenic potency as an agent for AMD treatment based on a mouse study [94]. Cabozntinib (Cabometyx^®^) was intravitreally applied into mice for anti-angiogenesis in a mouse model, and effectively down-regulated both VEGF receptor 2 activation and VEGF tyrosine kinase activation [95]. Sorafenib (Nexavar^®^) inhibits not only the activation of VEGFR-1 and VEGFR-2 but also the activation of VEGF receptor tyrosine kinase [96,97]. A biocompatibility study of the effects of multikinase inhibitors such as pazopanib, sorafenib (Nexavar^®^), and axitinib (Inlyta^®^) on human ocular cells found that, for all the inhibitors, there were no toxic effects until 7.5 μg/mL; when the cells were treated with 7.5 μg/mL or more of the inhibitors, the order of the effects on cell viability was pazopanib, sorafenib (Nexavar^®^), and axitinib (Inlyta^®^) [97]. SU11248/sunitinib (Sutent^®^) acts both as a VEGF receptor-2 inhibitor and a multitargeted tyrosine kinase receptor inhibitor, and its ability to treat AMD was confirmed based on a mouse model [98] and a chicken chorioallantoic membrane (CAM) model [99]. Table 6 shows a summary of the developed anti-VEGF drugs.

## 6. Drug Candidates Originated from Natural Products

Lutein and zeaxanthin exist as meso-zeaxanthin (a dipolar form, dihydroxylated carotenoid) in the retina, called macular xanthophylls, [100] which are included in dark green leafy vegetables such as kale, spinach, peas, etc. and egg yolks [101], and they suppress phototoxicity-induced oxidative stress and apoptosis in the visual system [102] and also induce G_2_/M phase arrest in the RPE [103]. Recently, there have been many trials that have investigated AMD drug candidates from natural products, with these products especially studied for their ability to regulate the RPE’s status (Table 7). These natural product candidates can be classified into (1) anti-apoptosis inducers, (2) cell cycle arrest modulators, and (3) VEGF inactivators. Based on the apoptotic pathway, the candidates can be grouped into (1) anti-oxidative stress, (2) anti-inflammation, and (3) anti-carbonyl stress candidates. One hundred percent EtOH extract of *Arctium lappa* L. leaves has been found to have plentiful amounts of phenolics and flavonoids, and it effectively suppressed both oxidative stress by scavenging effects of 2,2-diphenyl-1-picrylhydrazyl (DPPH) and 2,2′-azino-bis(3-ethylbenzothiazoline-6-sulfonic acid)(ABTS) and apoptosis by modulating bcl-2 family and caspase cascade [104]. Genipin is a glycosidic ligand that has been found to suppress H_2_O_2_-induced oxidative stress and apoptosis via nuclear factor-erythroid 2-related factor-2 (Nrf2), which regulates the expression of heme oxygenase-1 (HO-1) and NAD(P)H: quinine oxidoreductase 1 (NQOA) [105]. Delphinidin is one of the anthocyanins that exists in several fruits and vegetables [106] and has been found to significantly inhibit the synthesis of reactive oxygen species (ROS) through modulating antioxidative enzyme activation and apoptosis via regulating the bcl-2 family and caspase cascade [107]. Glabridin (isoflavonoid) from *Glycyrrhiza glabra* L. root has been demonstrated to inhibit oxidative stress and apoptosis through ERK1/2 and p38MAPK inactivation [108]. Wogonin (5,7-dihydroxy-8-methoxyflavone), isolated from *Scutellaria baicalensis* Georgi root, has been found to control LPS-induced inflammation changes such as IL-1β, IL-6, IL-8, TNF-α, COX-2, and inducible nitric oxide synthase (iNOS) via the Toll-like receptor 4 (TLR4)/NF-κB pathway [109]. Hot water extract of *Prunella vulgaris* var. L has been observed to prevent oxidative stress by ROS generation, carbonyl stress by MDA production, and inflammation through Nrf2/HO-1 pathway [110]. *Vaccinium uliginosum* L. hot water extract has been found to suppress A2E- and blue light-induced apoptosis by controlling caspase cascade and bcl-2 pathway [111]. Baicalin, a flavonoid isolated from *Scutellaria baicalensis* Georgi, has been found to inhibit Aβ-induced pyroptosis through modulating miR-223/NLRP3 inflammasome pathway [112]. β-carotene, as a typical carotenoid, has anti-oxidative effects and is one of the major components in tomatoes (*Lycopersicum esculentum* L.) [113] and has been reported to down-regulate H_2_O_2_-induced nitrotyrosine formation and protein carbonylation [38]. Bile acid is synthesized in the liver (primary bile acid) and in the colon (secondary bile acid), and is known as bile salt when it is conjugated with taurine or glycine [114]; the major bile salts include taurocholic acid, glycocholic acid, taurochenodeoxycholic acid, and glycochenodeoxycholic acid, etc. [115], and taurocholic acid, one of the major bile salts, has been demonstrated to inhibit angiogenesis-related cell proliferation, cell migration, and tube formation [116].

## 7. Perspectives

AMD, which was the third main cause of blindness in 2015 [3], is related to aging [1], and recently, its rate has increased considerably as birth rates have rapidly decreased. In 2017, the United Nations reported that the population of those 60 years old and over constituted 962 million people, more than twice (382 million) that recorded in 1980, with this demographic projected to be double (2.1 billion) the number recorded in 2017 [117] by 2050. In particular, the 60-year-old and over demographic’s share of the total population, which was about 20% in 2017, is expected to reach 35% in 2050. The prevalence of AMD-induced vision problems, therefore, is predicted to become much higher in the years to come. Along with aging, many factors can cause AMD, such as smoking, cataract surgery, BMI, cardiovascular problems, hypertension, fibrinogen, atherosclerosis, HDC-C, and the blue light of smart devices [9,10,11,12,13,14].

Although there are several medications for AMD, there are none that cure AMD perfectly without adverse effects. Device-based treatments, such as prophylactic laser therapy and photodynamic therapy, are very useful methods, as they involve noninvasive therapy, but, because these lasers use very high temperatures, if a prophylactic laser’s or photocoagulation laser’s curing point is wrong, these methods can damage the visual cells in the eye, such as macular edema, and lead to retinovascular diseases, among others [118,119]. Further, the most important problem of photodynamic therapy is its nonspecificity to cells, which means that photodynamic therapy can attack normal cells [120]. Therapies that use anti-inflammatory drugs such as corticosteroids or NAIDs are very useful for inhibiting inflammation and angiogenesis in AMD, but because corticosteroids have many adverse effects, such as hypertension, insulin resistance, insomnia, irritability, purpura, skin thinning, cataracts, glaucoma, gastric ulceration, etc. [121], their usage is limited to combined treatments with photodynamic therapy or anti-VEGF drugs; furthermore, as NSAIDs’ long-term therapeutic efficacy is unclear, more studies are needed [122]. Recently, anti-VEGF therapy has proven to be the most useful treatment method for AMD, as it has been found to inhibit angiogenesis from between one month and two months [85,90,91,92], but it has several adverse effects, such as bleeding and infection risk from intravitreal injection, the detachment of the retina, RPE tears, vitreoretinal fibrosis, etc. [123,124].

AMD is one of the incurable diseases; thus, it is most important to prevent its severity [1]. Although the prevalence rate of AMD increases with age, if we can prevent AMD from progressing to more severe forms, we can lower the population prevalence of blindness. From this reason, many natural products that can possibly constrain AMD severity have been used and investigated. Based on their modes of action, they can be divided into those that are preventers of RPE death/cell cycle arrest (anti-apoptosis inducers and cell cycle arrest modulators) [104,105,107,108,109,110,111,112] and those that are inhibitors of VEGF activation (VEGF inactivators) [38]. Natural products are safer than synthetic chemicals, as they have been used for a long time, they have an easy route of administration, with most able to be orally applied, and they can be used as combined therapies, such as with device-based therapy, anti-inflammatory drugs, and anti-VEGF molecules. We can conclude that AMD drug candidates from natural products can be applied to treat early and intermediate AMD [10]. 

## Figures and Tables

**Figure 1 ijms-22-11837-f001:**
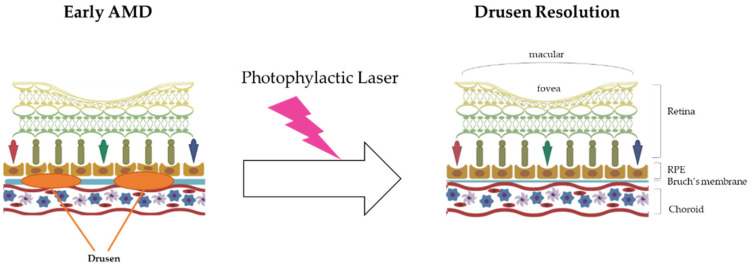
The mechanism of prophylactic laser therapy (photocoagulation therapy). In the early AMD stage, the drusen in Bruch’s membrane inhibit the blood supply to the RPE and eliminate the wastes near that area, such as lipofuscin (A2E). Finally, the accumulation of drusen in this area induces RPE death and worsens AMD. With energy (heat), prophylactic laser therapy resolves the build-up of drusen and then suppresses AMD progress.

**Figure 2 ijms-22-11837-f002:**
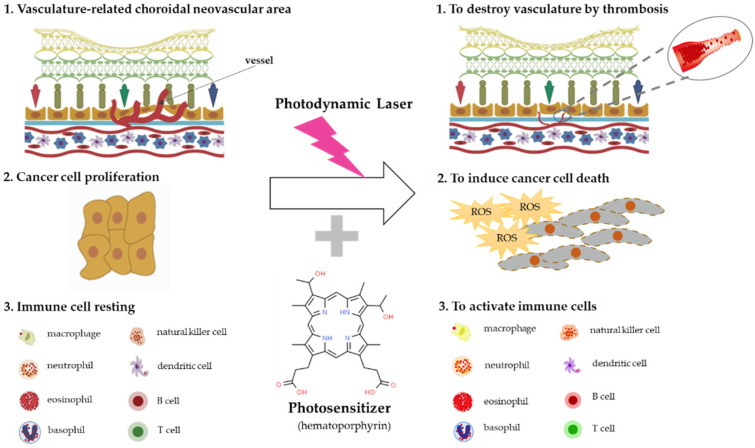
The mechanisms of photodynamic therapy. Photodynamic therapy generates reactive oxygen species (ROS) and destroys photosensitizer-binding fast proliferative cells, such as choroidal neovascularization (CNV) cells related to AMD via thrombosis and cancer cells through ROS synthesis, and stimulates immune cells’ activation.

**Figure 3 ijms-22-11837-f003:**
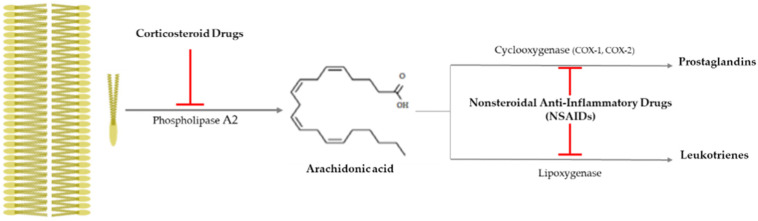
Anti-inflammatory pathway caused by corticosteroid drugs and nonsteroidal 
anti-inflammatory drugs (NSAIDs). Corticosteroid drugs inhibit the formation of 
arachidonic acid by causing plasma membrane breakage, which is caused by 
phospholipase A_2_. On the other hand, nonsteroidal anti-inflammatory 
drugs (NSAIDs) only control arachidonic acid and prostaglandins through 
cyclooxygenase 1 or 2 (COX-1 or -2). 
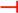
: to inhibit the follow action. COX, cyclooxygenase.

**Figure 4 ijms-22-11837-f004:**
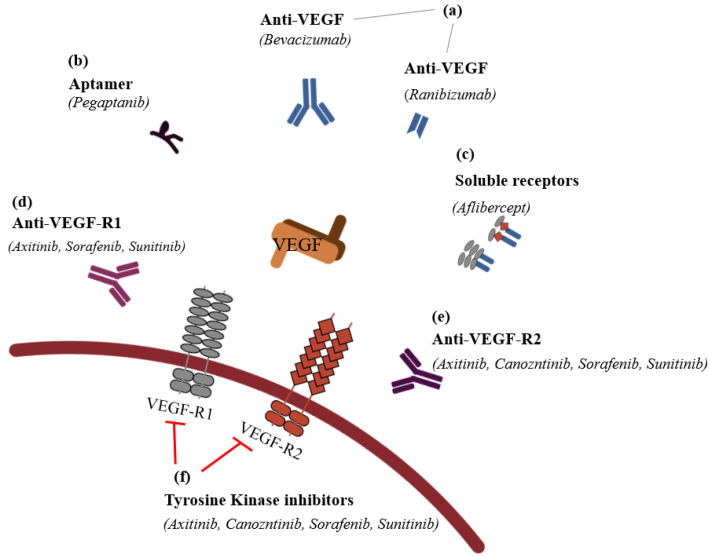
The classification of anti-vascular endothelial 
growth factors (modified from Noel et al., 2007 [84]). 
(**a**) VEGF antibodies directly 
bind the VEGF to inhibit its activation; these antibodies include humanized 
antibodies, such as bevacizumab (Avastin^TM^), and fragment of 
humanized antibodies, such as ranibizumab (Lucentis^TM^). (**b**) 
Apamers adhere on the VEGF to act like VEGF antibodies, and pegaptanib (Macugen^TM^) 
is the representative drug in this category. (**c**) In order to serve 
VEGF’s function, soluble receptors should bind on VEGF receptor-1 or -2, but if 
instead of binding VEGF receptors, VEGF adheres on exogeneous and artificial 
VEGF receptors (VEGF trap), such as aflibercept (Eylea^TM^), they 
cannot serve an angiogenesis function in the vascular endothelial cell. 
Anti-VEGF receptor 1 (**d**) and anti-VEGF receptor 2 (**e**) are 
receptors for VEGF, but if anti-VEGF-R1 and -R2 bind their receptors, VEGF 
cannot neovascularize. (**f**) If VEGFR tyrosine kinase inhibitors such as 
imatinib mesylate, sorafenib, sunitinib, and vatalanib inhibit the following 
process after VEGF’s binding, the CNV of VEGF receptors cannot occur. 
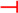
: to inhibit the follow action. VEGF, vascular endothelial growth factor; 
VEGF-R1, vascular endothelial growth factor receptor 1; VEGF-R2, vascular 
endothelial growth factor receptor 2.

**Table 1 ijms-22-11837-t001:** The categories of AMD medication.

Category	Description
Device-based treatment	(1) Prophylactic laser therapy (or photocoagulation therapy) is to eliminate the drusen. (2) Photodynamic therapy is to block the choroidal neovascularization simultaneously using a photosensitizer.
Anti-inflammatory drug treatment	(1) Corticosteroid drugs completely block the inflammatory pathway via phospholipase A2 inactivation. (2) Nonsteroidal anti-inflammatory drugs (NSAIDs) inhibit the activation of cyclooxygenase (COX-1, COX-2) to make prostaglandins (PGs) or/and lipoxygenase to make leukotrienes.
Anti-vascular endothelial growth factor (VEGF) treatment	(1) VEGF antibodies bind on VEGF and then inhibit VEGF activation. (2) Aptamer binds on VEGF in a manner similar to VEGF antibodies. (3) Soluble VEGF receptors bind on VEGF instead of VEGF receptors. (4) Anti-VEGF receptors (-1 or -2) bind VEGF receptors for blocking VEGF’s binding to VEGF receptors. (5) Tyrosine kinase inhibitors control the pathway following VEGF binding on VEGF receptors.
Nutritional treatment	According to the therapeutic mechanism, nutritional treatments can be classified below (1) Inhibition of oxidative stress and apoptosis; (2) Inhibition of inflammation and apoptosis; (3) Inhibition of oxidative stress, inflammation, and apoptosis; (4) Inhibition of apoptosis; (5) Inhibition of pyroptosis; (6) Inhibition of carbonyl stress; (7) Inhibition of G_2_/M phase arrest; (8) Inhibition of VEGF activation.

**Table 2 ijms-22-11837-t002:** The stage classification of age-related macular degeneration (AMD) depending on the drusen size and distribution area (from Klein 2014 [16]).

Category	Description
No AMD	No drusen, or questionable, small, or intermediate sized drusen (<125 μM in diameter) only, regardless of area of involvement, and no pigmentary abnormalities (defined as increased retinal pigment or RPE depigmentation present) OR no definite drusen with any pigmentary abnormality.
Mild early AMD	Small to intermediate sized drusen (<125 μM in diameter), regardless of area of involvement, with any pigmentary abnormality, OR large drusen (≥125 μM in diameter), with drusen area <331,820 μm^2^ (equivalent to O-2 circle, defined as a circle with diameter of 650 μm) and no pigmentary abnormalities.
Moderate early AMD	Large drusen (≥125 μM in diameter), with drusen area <331,820 μm^2^ and any pigmentary abnormality, OR large drusen (≥125 μM in diameter), with drusen area ≥331,820 μm^2^, with or without increased retinal pigment but with no RPE depigmentation.
Severe early AMD	Large drusen (≥125 μM in diameter), with drusen area ≥331,820 μm^2^ and RPE depigmentation present, with or without increased retinal pigment.
Late AMD	Pure geographic atrophy in the absence of exudative macular degeneration OR exudative macular degeneration, with or without geographic atrophy present.

AMD, age-related macular degeneration; RPE, retinal pigment epithelium.

**Table 3 ijms-22-11837-t003:** Summary of the prophylactic/photocoagulation laser therapeutics.

Classification of Laser	Advantage and Disadvantage	Reference
Xenon or Ruby in 1973	Resolves drusen but induces RPE death.	[49]
Argon Green (Low power) in 1998	Decreases RPE death but insufficiently.	[51]
Subthreshold diode in 2006	Decreases AMD progress but is not well established.	[52]
Nanosecond Pulse in 2011	Applies precise titration.	[53]

**Table 4 ijms-22-11837-t004:** Summary of anti-inflammatory drugs for AMD treatment.

Classification		Application Route and Therapeutic Effect	Reference
Corticosteroids	Dexamethasone	1. Topical application with artemisinin ‒ to control CNV in AMD progress	[68]
		2. Three combined therapies: dexamethasone, an anti-VEGF drug, and verteporfin with photodynamic therapy ‒ to decrease the anti-VEGF drug’s dosing quantity	[69]
	Triamcinolone Acetonide (TA)	1. Intravitreal injection ‒ to decrease the dosing time for macular edema and uveitis	[70]
		2. Combined intravitreal TA and bevacizumab injection ‒ to control CNV in AMD patients who failed to be treated by bevacizumab treatment alone	[71]
	Spironolactone (mineralocorticoid receptor antagonist)	Oral administration ‒ to suppress CNV in refractory AMD patients using intravitreal anti-VEGF injection	[72]
NSAIDs	Aspirin	Topical application ‒ to decrease AMD prevalence not only at high doses but also to have no effects at low doses	[73,74,75]
	Nepafenac (prodrug)	Topical application ‒ to induce potent anti-inflammatory activity, such as higher vascular permeability, longer inhibition of PGE synthesis, and better corneal penetration	[76]
	Diclofenac, Ketorolac	Intravitreal diclofenac and ketorolac injection ‒ to control inflammation in an LPS-induced inflammation rabbit model	[77]

NSAIDs, nonsteroidal anti-inflammatory drugs; CNV, choroidal neovascularization; AMD, age-related macular degeneration; VEGF, vascular endothelial growth factor; TA, triamcinolone acetonide; LPS, lipopolysaccharide.

**Table 5 ijms-22-11837-t005:** Summary of the currently used anti-vascular endothelial growth factor (anti-VEGF) drugs for AMD treatment.

Classification		Inhibits	Half-Life	Doses	Reference
VEGF Antibody	Humanized Ab: bevacizumab (Avastin^TM^; Genentech, South San Francisco, CA, USA)	All VEGF-A isoforms	4~5 days	1.25 mg/month	[85]
	Fragment Humanized Ab: Ranibizumab (Lucentis^TM^; Genentech, South San Francisco, CA, USA)	All VEGF-A isoforms	6~10 days	0.5 mg/month	[85]
Aptamer	Pegaptanib (Macugen^TM^; Pfizer, New York, NY, USA)	VEGF-A_165_ isoform	8~14 days	0.3 mg/two months	[90]
Soluble Receptor	Aflibercept (Eylea^TM^; Regeneron Pharmaceuticals Inc., Tarrytown, NY, USA)	All VEGF-A isoforms, VEGF-B, PGF	5~6 days	2 mg/two month	[91]

VEGF, vascular endothelial growth factor; PGF, placental growth factor.

**Table 6 ijms-22-11837-t006:** Summary of the developed anti-vascular endothelial growth factor (anti-VEGF) drugs for AMD treatment.

Agent	Inhibition Point (Mode of Action)			Application			References
	VEGF R-1	VEGF R-2	Tyrosine Kinase	Model	Route	Dose	
Axitinib (Inlyta^®^; Pfizer, Inc, New York, NY. USA)	◯	◯	◯	Mouse	P.O.	5 mg/kg	[94]
Sorafenib (Nexavar^®^; LC Laboratories, Woburn, MA, USA)	◯	◯	◯	RPE cell	Media	1 μg/mL	[96,97]
Sunitinib or SU11248 (Sutent^®^; Pfizer Inc., New York, NY, USA)	◯	◯	◯	Mouse	P.O.	80 mg/kg	[98]
				Chicken chorioallantoic membrane (CAM)	Topical	20 μL of 20 μM	[99]
Cabozntinib (Cabometyx^®^; Exelixis Pharmaceuticals, Inc., Alameda, CA, USA)	-	◯	◯	Mouse	Intravitreal	2 μg/head	[95]

**Table 7 ijms-22-11837-t007:** Summary of AMD drug candidates developed from natural products.

Therapeutic Mechanism	Natural Product			Application			References
	Species	Applied Characteristic	Effective Compound	Model	Route	Minimum Effective Dose	
Inhibition of oxidative stress and apoptosis	*Arctium lappa* L. leaf	100% EtOH extract	Phenolic and flavonoid	RPE cell	Media	30 μg/mL for 24 h	[104]
				Mouse	I.P.	50 mg/kg for 4 w	
	*Eucommia ulmoides*	Genipin (glycosidic ligand)	Genipin (glycosidic ligand)	ARPE-19 cell	Media	30 μM for 24 h	[105]
	Fruit or Vegetable	Delphinidin (anthocyanidin)	Delphinidin (anthocyanidin)	ARPE-19 cell	Media	25 μg/mL for 24 h	[107]
	*Glycyrrhiza glabra* L. root	Glabridin (isoflavonoid)	Glabridin (isoflavonoid)	RPE cell	Media	2 μM for 2 h	[108]
				Mouse	I.P.	20 mg/kg for 1 w	
Inhibition of inflammation and apoptosis	*Scutellaria baicalensis* Georgi root	5,7-dihydroxy-8-methoxyflavone (wogonin)	5,7-dihydroxy-8-methoxyflavone (wogonin)	ARPE-19 cell	Media	10 μM for 24 h	[109]
Inhibition of oxidative stress, inflammation, and apoptosis	*Prunella vulgaris* var. L	Water extract	Rosmarinic Acid	ARPE-19 cell	Media	100 μg/mL for 24 h	[110]
				Mouse	P.O.	100 mg/kg for 4 day	
Inhibition of apoptosis	*Vaccinium uliginosum* L	Water extract	Polyphenol	ARPE-19 cell	Media	100 μg/mL for 24 h	[111]
Inhibition of pyroptosis	*Scutellaria baicalensis* Georgi	Baicalin	Baicalin	ARPE-19 cell	Media	50 μg/mL for 72 h	[112]
Inhibition of carbonyl stress	*Lycopersicum esculentum* L. (Tomato)	n-hexane extract	β-carotene	ARPE-19 cell	Media	1 μM-β-carotene for 24 h	[38]
Inhibition of G_2_/M phase arrest	Fruit or Vegetable	Lutein	Lutein	ARPE-19 cell	Media	25 μg/mL for 24 h	[103]
Inhibition of VEGF activation	Bile Acid (Animal)	Taurocholic acid	Taurocholic acid	HRPEpiC cell	Media	100 μM for 48 h	[116]
				RF/6A cell			

RPE, retinal pigment epithelium; I.P., intraperitoneal injection; w, week; EtOH, ethanol; ARPE-19, adult retinal pigment epithelial cell line-19; P.O., oral administration; HRPEpiC, human retinal pigment epithelial cells; RF/6A, chorioretinal cells from *Rhesus macaque* (non-human primate).

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
