# Peer review of "Medication Trends for Age-Related Macular Degeneration"

_ijms, 2021, doi:10.3390/ijms222111837_

Round 1

Reviewer 1 Report

The manuscript by Cho et al. represents a review article summarizing the current AMD medications, their advantages and disadvantages considering how to prevent or improve the symptoms of intermediate AMD. The manuscript is timely, is well written, well illustrated and easy to follow. Nonetheless I have some suggestions, which I hove would help the authors improving the manuscript.

- Due to the fact that the major focus of this review is pointed on the current medications, current title is misleading. On the other hand, an addition of a chapter focusing on the novel types of AMD therapeutic candidates would cut out this gap. In my opinion, ‘Perspectives’ chapter should contain the authors’ opinion on the most perspective areas of novel medications as well.

Minor:

- Introduction could be improved by the addition of schemes/tables illustrating classifications;

- Figure 4 legend contains ‘(1)’, ‘(2)’ etc., while the Figure ‘(a)’, ‘(b)’ etc.  

Author Response

Reviewer #1

The manuscript by Cho et al. represents a review article summarizing the current AMD medications, their advantages and disadvantages considering how to prevent or improve the symptoms of intermediate AMD. The manuscript is timely, is well written, well illustrated and easy to follow. Nonetheless I have some suggestions, which I hove would help the authors improving the manuscript.

- Due to the fact that the major focus of this review is pointed on the current medications, current title is misleading. On the other hand, an addition of a chapter focusing on the novel types of AMD therapeutic candidates would cut out this gap.

Ans) Thank you so much for informative suggestion based on your completely understanding on my manuscript. I deeply considered your comments which were controversially suggested and I concluded that I omitted the word, ‘current’. In this review I hope that we could provide the information about how to treat AMD patients. The medication for AMD patients could be classified according to the grade (severity) of AMD, the method to treat AMD, the therapeutic mechanism, etc. In this review I hope that the readers on my review could get the information of the medication for AMD including the knowledge of advantage and disadvantage on each method. Finally, if I omit the word, ‘current’ it would be easy to understand the purpose of this review.

 In my opinion, ‘Perspectives’ chapter should contain the authors’ opinion on the most perspective areas of novel medications as well.

Ans) Thank you so much for informative comment and I added the several sentences in the Perspectives section.

Minor:

- Introduction could be improved by the addition of schemes/tables illustrating classifications;

Ans) Thank you so much for the comment and I added the Table 1 for the category of AMD medication.

- Figure 4 legend contains ‘(1)’, ‘(2)’ etc., while the Figure ‘(a)’, ‘(b)’ etc.  

Ans) Thank you so much for the generous comment and I amended them.

Reviewer 2 Report

Dear Author,  I read with interest your manuscript. I think that many issues have to be clarified before it could be considered for publication.   The definition and subtypes of the disease are not correct, the late form is not only neovascular. Clinical features cause several clinical signs and symptoms that must be enumeratted, in particular it causes central blindness and not a complete blindness. Considering avaialble therapies, laser photocoagulation and photo dynamic therapies stil represent the past and nowadys are completely stopped in particular for the neovascular progression. Antiinflammatory drugs (steroids and FANS) may represent only adjunctive elements in the gold standard therapy for  neovascular form. For the intermediate and early form actually we have only drugs that reduce the oxidative and inflammatory substrate. It's important to cite the results of the AREDS study in its different publications. It underlines the value of selected basic factor useful to slow the progression of the pathology. I appreciate the list of other factors you showed but remember that very few clinical data are available on early AMD treatment.  If you declare current trends on the title you may consider the effective clinical application of these substances. You may underline the difficulties to evaluate the clinical value of these product, due to the slow chronicity of the pathology, the lack of significative and effective instrumental devices to measure the progression of oxidative damage,. etc... In conclusion your sentence about "the need of treatment without adverse events" express a correct need but we are actually far from that and no clinical results are available.     Best regards

Author Response

Reviewer #2

Dear Author,  I read with interest your manuscript. I think that many issues have to be clarified before it could be considered for publication.

The definition and subtypes of the disease are not correct, the late form is not only neovascular. Clinical features cause several clinical signs and symptoms that must be enumeratted, in particular it causes central blindness and not a complete blindness.

Ans) Thank you so much for the comments and I amended them.

Considering avaialble therapies, laser photocoagulation and photo dynamic therapies stil represent the past and nowadys are completely stopped in particular for the neovascular progression.

Ans) I completely agree with the comment.

Antiinflammatory drugs (steroids and FANS) may represent only adjunctive elements in the gold standard therapy for  neovascular form. For the intermediate and early form actually we have only drugs that reduce the oxidative and inflammatory substrate. It's important to cite the results of the AREDS study in its different publications.

Ans) Thank you so much for the informative comments and I referred to many publications-related with AREDS.

It underlines the value of selected basic factor useful to slow the progression of the pathology. I appreciate the list of other factors you showed but remember that very few clinical data are available on early AMD treatment. 

Ans) Thank you so much for the informative comments.

If you declare current trends on the title you may consider the effective clinical application of these substances. You may underline the difficulties to evaluate the clinical value of these product, due to the slow chronicity of the pathology, the lack of significative and effective instrumental devices to measure the progression of oxidative damage,. etc...

Ans) Thank you so much for informative suggestion based on your completely understanding on my manuscript. I deeply considered your comments which were controversially suggested and I concluded that I omitted the word, ‘current’.

In conclusion your sentence about "the need of treatment without adverse events" express a correct need but we are actually far from that and no clinical results are available.    

Ans) Thank you so much for the comment and I amended several sentences in the Perspective section.

Best regards

Round 2

Reviewer 1 Report

my concerns have been addressed, thank you. 

Reviewer 2 Report

Thank you or your revision.

I appreciated the modifications on it

Best regards